# Principled and Tractable RL for Reasoning with Diffusion Language Models

## Abstract

Diffusion large language models (dLLMs) are a new paradigm of non-autoregressive language models that are trained to predict multiple tokens in parallel and generate text via iterative unmasking. Recent works have successfully pretrained dLLMs to parity with autoregressive LLMs at the 8B scale, but dLLMs have yet to benefit from modern post-training techniques, e.g. reinforcement learning (RL), that have proven effective for autoregressive models. Crucially, current algorithms aren't directly compatible with diffusion models due to their lack of left-to-right sequence likelihood factorization. Moreover, existing attempts at dLLM post-training with RL rely on unprincipled heuristics such as mean-field approximations. In this work, we present Amortized Group Relative Policy Optimization (AGRPO), an on-policy RL algorithm designed specifically for dLLMs. Our key insight is that by casting the denoising process as a multi-step Markov decision process, we can use Monte Carlo sampling to compute an unbiased policy gradient estimate, making AGRPO the first tractable yet faithful adaptation of policy gradient methods for dLLMs. We demonstrate AGRPO's effectiveness on different math/reasoning tasks, achieving up to +10.0% absolute gain on GSM8K, 3.8x performance on the Countdown task over the baseline LLaDA model, and 3.4x performance gains over comparable RL methods such as diffu-GRPO. Furthermore, these gains persist across different numbers of sampling steps at inference time, achieving better tradeoffs between compute and performance. Our results establish that online RL algorithms can be extended to diffusion LLMs in principled ways, maintaining both theoretical soundness and practical effectiveness.

## 1 Introduction

Many recent efforts in LLM research have centered around reinforcement learning, specifically in the verifiable reward (RLVR) setting. In a typical setup, base models are trained on math or coding problems and incentivized to reason through the solution step-by-step, getting a reward if the final answer is correct. The main goal of RLVR is to elicit mathematical thinking/reasoning capabilities, allowing models to solve complex real-world tasks.

This wave of interest in RL and reasoning, initially spurred by models like OpenAI's o1 (OpenAI et al., 2024) and DeepSeek's R1 (DeepSeek-AI et al., 2025), has led to the development of numerous RL algorithms designed specifically for transformer-based autoregressive (AR) LLMs. With the success of these algorithms, among them Group Relative Policy Optimization (GRPO) (Shao et al., 2024), AR LLMs have grown incredibly strong on problem-solving benchmarks, often matching or exceeding human expert level. Closed frontier models have even achieved gold medal performance at competitions such as the IMO and IOI, a remarkable feat (Luong & Lockhart, 2025; Lin & Cheng, 2025).

In a parallel line of research, dLLMs have recently emerged as an alternative to the traditional autoregressive paradigm. Continuous diffusion models have long been established as the dominant framework for image and video generation, relying on a denoising/score matching objective. Works such as D3PM (Austin et al., 2021) and SEDD (Lou et al., 2024) successfully transferred this diffusion framework to discrete settings, including language. Successive efforts such as MDLM (Sahoo

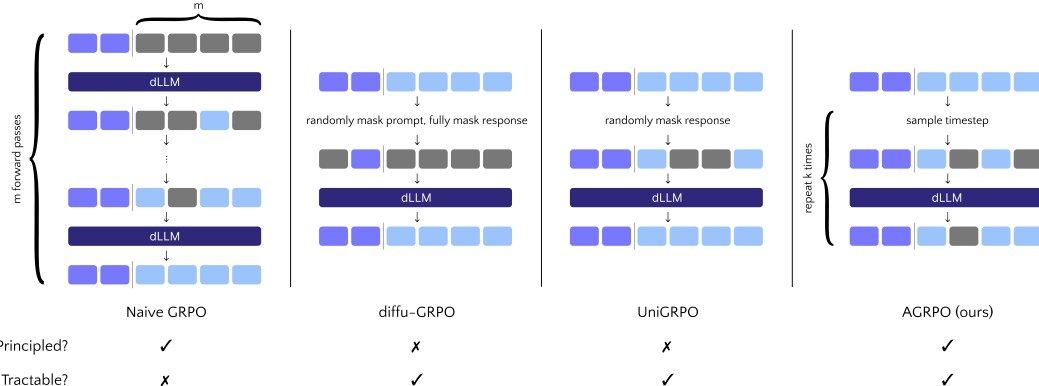

Figure 1: A comparison of different RL post-training algorithms for dLLMs. Existing algorithms designed for traditional LLMs such as GRPO would entail $O(\text{response length})$ forward passes for dLLMs. Current tractable techniques for dLLMs involve heuristic approximations, resulting in biased policy gradients. Our proposed algorithm takes a different approach that remains tractable and faithful to the original GRPO objective.

et al., 2024) and RADD (Ou et al., 2025) have simplified and merged different formulations, with consensus settling on a masked diffusion framework.

While autoregressive models are trained to predict the next token in left-to-right order, dLLMs are trained to predict an *arbitrary* number of tokens in *arbitrary* positions. Current state-of-the-art dLLMs, such as LLaDA-8B (Nie et al., 2025) and MMaDA (Yang et al., 2025), are close to or on par with open-source AR models such as LLaMA3-8B and Qwen2.5-7B on common NLP benchmarks.

Once trained, these models can generate text with greater flexibility than standard left-to-right prompting (e.g. infilling) by starting with a completely or partially masked sequence and iteratively unmasking tokens. In addition, dLLMs have the benefit of being able to trade off compute and quality at inference time: one can decrease the number of steps/forward passes to save compute at the cost of slightly worse generations (Lou et al., 2024). However, these models still struggle to match AR models in downstream tasks that require long-form thinking and reasoning. This discrepancy in post-training stems from fundamental challenges in designing training objectives for dLLMs: AR models have easy access to token- and sequence-level likelihoods through AR factorization, whereas diffusion models must resort to approximations or ELBO-like bounds on likelihood. Unlocking true reasoning capabilities would be a giant leap forward for dLLMs, solidifying them as a true rival of AR LLMs.

Our work helps dLLMs close this gap by proposing a principled, unbiased form of GRPO designed especially for dLLMs: Amortized GRPO (AGRPO). Unlike previous work, we first establish a multi-step MDP formulation of the post-training problem which allows for exact action likelihoods. Then, through a simple modification of the original GPRO objective — by viewing the inner sum over all tokens as an expectation over timesteps — we show how to make training tractable for dLLMs.

Our main contributions are as follows:

- **Soundness.** Building off of GRPO, we derive a theoretically sound policy gradient objective that takes into account the multi-step denoising nature of dLLMs. We show how to efficiently compute it and also explain why comparable methods, based on heuristic approximations of sequence likelihoods, fail to be theoretically sound.

- **Efficiency.** We explain how to practically implement our proposed algorithm in a way that is both memory- and compute-efficient, and discuss various tradeoffs surrounding implementation.

- **Efficacy.** We train models using AGRPO on three mathematical reasoning tasks: GSM8K, MATH, and Countdown, achieving absolute gains of up to +10.0%, +6.6%, and +29.6%, respectively. In addition to outperforming previous approximation-based methods, we show

that models trained with AGRPO consistently retain high accuracy when evaluated with different numbers of sampling steps, a previously unexplored aspect of dLLM post-training.

# 2 UNDERSTANDING DIFFUSION LANGUAGE MODELS

## 2.1 PRETRAINING

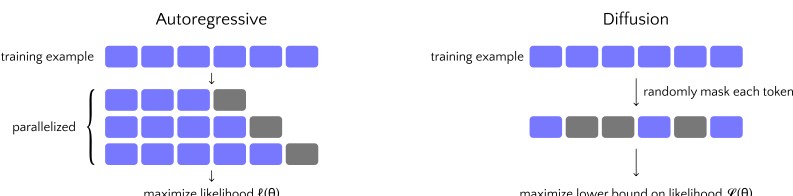

Figure 2: Models under the autoregressive and diffusion paradigms are trained on different objectives. Next-token prediction (left) is a narrower, easily parallelizable task, whereas masked token prediction (right) is harder and less conducive to parallelism since it involves predicting multiple tokens. Diffusion models must also optimize for a lower bound rather than the exact likelihood.

The most common form of discrete diffusion for language is the masked (or "absorbing") approach, where models are trained to reverse data corrupted by randomly masking tokens (Lou et al., 2024; Sahoo et al., 2024; Arriola et al., 2025). Concretely, given a distribution $p$ on sequences of discrete tokens $x = (x_1, \ldots, x_n)$, models are trained to maximize the following evidence-based lower bound (ELBO) on the likelihood (Nie et al., 2025; Ou et al., 2025):

$$\mathcal{L}(\theta) = \mathbb{E}_{\substack{t \sim U[0,1] \\ x, x^t \sim p^t}} \left[ \frac{1}{t} \sum_{x_i^t = \blacksquare} \log p_\theta(x_i \mid x^t) \right] \tag{1}$$

where $x, x^t \sim p^t$ means that $x$ is sampled from $p$ and $x^t$ is obtained from $x$ by independently setting each token $x_i$ to the mask token $\blacksquare$ with probability $t$.[1] This masking process is also known as the "forward process." For example, $x^t$ might have 3 $\blacksquare$'s after the forward process, so the objective would be a sum of 3 cross-entropy terms. Similar to BERT (Devlin et al., 2019), the goal is for the model to learn conditional marginal distributions of masked tokens given their context.

A crucial point is that compared to the classic AR objective, this masked token prediction objective is harder/more general, since the unmasking order can be arbitrary and the model must predict *all* masked tokens. By contrast, AR models constrain themselves to modeling the *next* token in left-to-right order. The benefits of imposing this causal constraint are twofold: it lets training be parallelized so that transformer models can learn to generate the whole example with a single forward pass, and it also lets AR models maximize the exact likelihood (via the chain rule) rather than a lower bound (see Figure 2).

## 2.2 INFERENCE

To generate text, dLLMs start with an all- or partially-masked sequence, obtain marginal distributions for each masked token, and then unmask some of these by sampling from their marginals. The tokens to be unmasked can be chosen either randomly, adhering to the theoretical "backward process," or by keeping the tokens with highest probability, as proposed by Nie et al. (2025). (We refer to these as "random" and "low-confidence" unmasking, respectively.) The rest of the tokens are kept the same, and this new sequence is fed back into the model; this process is repeated until all tokens are unmasked.

Throughout this paper, we use $m$ to refer to the number of sampling steps, and $n$ to refer to the sequence length. Note that unlike autoregressive models, the ratio $n/m$ can be adjusted at inference time, a nice advantage for diffusion models. Typically, the number of tokens unmasked at each

---

[1]One can derive an equivalent discrete ELBO that samples over $x^t$ with exactly $j$ $\blacksquare$'s, summing from $j = 1$ to $n$, which is empirically more stable (Ou et al., 2025; Zheng et al., 2025).

step $n/m$ is chosen to be relatively small ($\leq 8$) — unmasking higher tokens at each step severely degrades quality as measured by perplexity/accuracy (Lou et al., 2024; Nie et al., 2025). We show in later sections that for specific problem-solving tasks, post-training actually allows for much higher values of $n/m$ without clear degradation.

For more details on dLLM inference, see Appendix F.

## 3 FROM PPO TO AGRPO

### 3.1 RL AS A MULTI-STEP MDP

Markov decision processes (MDPs) are a formalization of sequential decision-making problems consisting of a state space $S$, an action space $A$, a transition kernel $P(\cdot \mid s, a)$, and a reward function $r(s, a)$. In the context of LLM post-training, $S$ corresponds to the context (i.e. the prompt), $A$ corresponds to tokens, and $r$ is provided via an external model or a ground truth answer. (Transitions are deterministic, so $P$ is irrelevant.)

Thanks to AR factorization, one can view AR LLMs as policies that generate distributions over the entire sequence, i.e. a *one-step* MDP. However, diffusion models don't admit such a clean factorization, which makes sequence likelihoods intractable (hence the ELBO training objective). Instead, it is more natural to consider a *multi-step* MDP formulation where an action corresponds a single unmasking step, which directly aligns with how diffusion models are parameterized. This formulation has been adopted by previous works on RL for diffusion such as DDPO and Flow-GRPO Black et al. (2024); Liu et al. (2025a) and crucially allows for *exact likelihoods*, sidestepping the need for sequence-level lower bounds or approximations.

Note that although this multi-step perspective allows for per-action likelihoods, computing them naively would require a separate forward pass for every unmasking step, making it impractical. We show how AGRPO addresses this problem in section 4.1.

### 3.2 POLICY GRADIENT METHODS FOR LANGUAGE MODELS

Early attempts at RL with LLMs (Ouyang et al., 2022; Ziegler et al., 2020) focused on alignment with human preferences using standard policy gradient methods such as Proximal Policy Optimization (PPO). PPO involves training separate models for the reward $r$ and value $V$, which are combined with Generalized Advantage Estimation (GAE) to produce advantage estimates $A_t$ (Schulman et al., 2017), a potentially cumbersome step for larger models.

More recently, RL efforts have focused on reasoning capabilities, specifically for domains with verifiable rewards such as math and coding. Notably, models trained with PPO-like algorithms improve reasoning abilities beyond SFT Luo et al. (2025), and even demonstrate emergent capabilities such as backtracking and self-correction (Xiong et al., 2025).

A nice property of RLVR environments is that they obviate the need for the external reward model in PPO. Group Relative Policy Optimization (GRPO), a widely used algorithm proposed by Shao et al. (2024), goes further by replacing the value model with the mean reward across a group of rollouts. With this setup, the complexity and computational burden of training is greatly reduced, making it a dominant choice. Models trained with GRPO showed much stronger performance across math benchmarks and even exhibited "A-ha moments" indicative of genuine mathematical reasoning (DeepSeek-AI et al., 2025).

### 3.3 THE GRPO OBJECTIVE

**Notation.** Let $D$ be a distribution over questions $q$, $\{o^i\}_{i=1}^G$ a group of $G$ outputs (or rollouts) generated from policy $\pi_{old}$ (which can optionally be offline), $r_i$ the respective rewards, and $A_i = \frac{r_i - \text{mean}\{r_i\}}{\text{std}\{r_i\}}$ the (normalized) advantage estimates. We use $o_t$ to denote the intermediate state of rollout $o$ at timestep $t$. For AR models, this is just the first $t$ tokens; for diffusion models, this is the partially masked state after the $t$-th denoising step (so $o_0$ is the fully masked state).

Omitting the $\min$ and $D_{\text{KL}}$ terms for clarity, GRPO's objective function is

$$\mathcal{J}(\theta) = \mathbb{E}_{\substack{q \sim D \\ \{o^i\}_{i=1}^G \sim \pi_{old}(\cdot|q)}} \left[ \frac{1}{G} \sum_{i=1}^{G} \frac{1}{|o^i|} \sum_{t=1}^{|o^i|} \text{clip} \left( \frac{\pi_\theta(o_t^i \mid q, o_{t-1}^i)}{\pi_{old}(o_t^i \mid q, o_{t-1}^i)}, 1 - \varepsilon, 1 + \varepsilon \right) A_i \right]. \quad (2)$$

This form is derived from the one-step MDP view and naturally lends itself to AR LLMs, where all token likelihoods can be exactly computed in a single forward pass thanks to AR factorization. However, dLLMs do not admit a clean sequence-level factorization, which makes this objective infeasible. Prior works work around this by optimizing some approximation or lower bound instead, as we discuss in the next section.

With the multi-step MDP view, we can get the exact token likelihoods, but computing the entire objective would require $O(|o^i|)$ forward passes (one per inner term), which is simply intractable for online RL. We propose a novel way of addressing this problem via timestep sampling in Section 4.1.

## 3.4 PREVIOUS ATTEMPTS AT dLLM POST-TRAINING

Although dLLM post-training progress remains relatively unexplored, there have been several attempts to adapt RL techniques such as GRPO. Here we highlight two recent such attempts and explain why they fail to hold up under theoretical analysis.

**diffu-GRPO.** Zhao et al. (2025) propose a two-stage dLLM post-training process consisting of SFT on reasoning traces followed by RL. We focus on their proposed RL algorithm, diffu-GRPO, which assumes the one-step MDP view. diffu-GRPO uses a mean-field approximation $\log \pi_\theta(o|q) \approx \sum_t \log \pi_\theta(o_t|q)$ and estimates all token likelihoods $\pi_\theta(o_t|q)$ under a *single* denoising step, i.e. from a fully masked state. They also introduce random masking to the prompt (Figure 1) in a manner similar to dLLM pretraining, which is claimed to stabilize training. However, there is an evident mismatch with respect to the RL environment: rollouts are generated with *multiple* denoising steps, and the context/partially masked state clearly matters when the model is unmasking tokens (i.e. choosing actions).

**UniGRPO.** MMaDA is a multimodal dLLM developed by Yang et al. (2025) which is post-trained by UniGRPO, an algorithm that unifies RL training across different tasks and modalities. Identifying the lack of different masking levels in diffu-GRPO as a potential flaw, they propose randomly masking the entire sequence multiple times and approximating token likelihoods by averaging across samples where that token is masked. More formally, given rollout $o$ and $\alpha \in [0, 1]$, let $o^\alpha$ denote a randomly noised version of $o$ where each token is independently masked with probability $\alpha$ (as in Section 2.1). UniGRPO optimizes an ELBO-like bound

$$\mathbb{E}_{\alpha \sim U[0,1]} \left[ \frac{1}{M} \sum_{o_j^\alpha = \blacksquare} \log \pi_\theta(o_j \mid q, o^\alpha) \right]$$

where $M = \sum_j \mathbf{1}_{\{o_j^\alpha = \blacksquare\}}$ is the number of mask tokens and the RHS is estimated by sampling $\alpha$.[2] This follows a denoising heuristic where the model learns to recover the true output from randomly perturbed context, similar to pretraining.

*Remark.* The GRPO objective also includes a KL penalty $D_{\text{KL}}(\pi_\theta || \pi_{\text{ref}})$ term which diffu-GRPO and UniGRPO both interpret as a sequence-level KL divergence, following the one-step MDP view. Since the $D_{\text{KL}}$ term requires exact likelihoods, this must also be approximated for diffusion models.

Although both methods proposed above obtained positive empirical results, they both suffer from a reliance on one-step likelihood approximations. These approaches make RL tractable, but come at the cost of theoretical inconsistency. A better approach would take into account the multi-step framing of diffusion policies (Section 3.1), allowing for exact action probabilities, while remaining tractable.

With these considerations in mind, we propose a new algorithm called Amortized Group Relative Policy Optimization (AGRPO), the first principled adaptation of policy gradient methods to dLLMs

---

[2]To ensure every token is masked at least once, the algorithm enforces $\alpha = 1$ for the last sample. This breaks theoretical soundness since samples are no longer distributed $\sim U[0,1]$.

that computes an *unbiased* gradient estimate without any reliance on likelihood bounds or approximations.

# 4 AMORTIZED GRPO

## 4.1 DERIVING THE AGRPO OBJECTIVE

In this section, we show how to reinterpret the GRPO objective as an expectation across timesteps with respect to the uniform measure. Assume the number of sampling steps per rollout is constant $|o^i| = m$ (since dLLMs must fix this number before generation).[3] For a given prompt, consider the following form of equation 2, with clipping removed for clarity:

$$\mathcal{J}(\theta) = \frac{1}{G} \sum_{i=1}^{G} \frac{1}{m} \sum_{t=1}^{m} \frac{\pi_\theta(o_t^i \mid q, o_{t-1}^i)}{\pi_{old}(o_t^i \mid q, o_{t-1}^i)} A_i = \frac{1}{G} \sum_{i=1}^{G} \mathbb{E}_{t \sim \{1, \dots, m\}} \left[ \frac{\pi_\theta(o_t^i \mid q, o_{t-1}^i)}{\pi_{old}(o_t^i \mid q, o_{t-1}^i)} A_i \right]. \quad (3)$$

The RHS can now be estimated via Monte Carlo (MC) sampling by drawing $k \ll m$ timesteps from the uniform distribution on $\{1, \dots, m\}$, computing the *exact* action likelihoods, and averaging (Figure 1).

Following subsequent improvements to GRPO, such as Dr. GRPO (Liu et al., 2025b), we use unnormalized advantages $A_i = r_i - \text{mean}\{r_i\}$. This helps avoid bias from particularly easy or hard problems where advantages have low variance.

The full AGRPO objective is

$$\mathcal{J}(\theta) = \mathbb{E}_{\substack{q \sim D \\ \{o^i\}_{i=1}^{G} \sim \pi_{old}(\cdot \mid q)}}$$

$$\left[ \frac{1}{G} \sum_{i=1}^{G} \mathbb{E}_{t \sim \{1, \dots, m\}} \left[ \min \left( \rho_t^i A_i, \text{clip} \left( \rho_t^i, 1 - \varepsilon, 1 + \varepsilon \right) A_i \right) - \beta D_{\text{KL}}(\pi_\theta \,\|\, \pi_{\text{ref}}) \right] \right] \quad (4)$$

where

$$\rho_t^i = \frac{\pi_\theta(o_t^i \mid q, o_{t-1}^i)}{\pi_{old}(o_t^i \mid q, o_{t-1}^i)}$$

and the KL term represents the divergence at timestep $t$, i.e. $D_{\text{KL}}(\pi_\theta(\cdot \mid q, o_{t-1}^i) \,\|\, \pi_{\text{ref}}(\cdot \mid q, o_{t-1}^i))$. Since we adopt a multi-step framing where actions are single denoising steps, the KL term is over the specific tokens unmasked at $t$, so we can estimate it via MC sampling without relying on sequence-level approximations. We use Schulman (2020)'s unbiased KL estimator

$$D_{\text{KL}}(p\|q) = \mathbb{E}_{x \sim p} \left[ \frac{q(x)}{p(x)} - \log \frac{q(x)}{p(x)} - 1 \right]$$

which is also used by the original GRPO paper (Shao et al., 2024).

Algorithm 1 provides an overview of our proposed algorithm. For practical considerations, including how to efficiently retrieve partially masked states $o_t$ and compute gradients, see Appendix E.

*Remark.* Although our algorithm includes $\mu$ as a hyperparameter, for this paper we adopt the common practice of assuming $\mu = 1$, so the algorithm is fully on-policy (Liu et al., 2025a).

## 4.2 LOW-DISCREPANCY SAMPLING

Once $k$ is fixed, a naive MC sampling algorithm would draw $k$ i.i.d. samples and average them. However, we can actually reduce the variance further by introducing correlation across samples so that they collectively "cover" a wide range of timesteps while ensuring that the marginal distribution for each sample is still uniform on $\{1, \dots, m\}$. This is known as low-discrepancy sampling, and is used in practice to lower training variance for both continuous and discrete diffusion models (Kingma et al., 2021; Sahoo et al., 2024; Zheng et al., 2025). We follow Zheng et al. (2025)'s discrete low-discrepancy sampler, which is detailed in Appendix D.

---

[3] Liu et al. (2025b) show that even with AR models, replacing the $\frac{1}{|o^i|}$ factor by a constant $\frac{1}{C}$ helps mitigate response length bias.

---

**Algorithm 1** Amortized Group Relative Policy Optimization (AGRPO)

---

**Require:** policy $\pi_\theta$, # sampling steps $m$, # MC samples $k$

  $\pi_{\text{ref}} \leftarrow \pi_\theta$
  **while** not converged **do**
    $\pi_{old} \leftarrow \pi_\theta$
    sample prompt $q$
    sample rollouts $\{o^i\}_{i=1}^G \sim \pi_{old}(\cdot \mid q)$
    compute advantages $\{A_i\}$
    **for** $\ell = 1$ **to** $\mu$ **do**
      **for** $j = 1$ **to** $k$ **do**
        select $t \in \{1, \ldots, m\}$ uniformly with low-discrepancy sampler (Section 4.2)
        $\rho_t^i \leftarrow \frac{\pi_\theta(o_t^i|q,o_{t-1}^i)}{\pi_{old}(o_t^i|q,o_{t-1}^i)}$
        $M_j \leftarrow \sum_{i=1}^G \left[ \min\left(\rho_t^i A_i, \text{clip}\left(\rho_t^i, 1-\varepsilon, 1+\varepsilon\right) A_i\right) - \beta D_{\text{KL}}(\pi_\theta \,\|\, \pi_{\text{ref}}) \right]$
      **end for**
      compute AGRPO objective $\mathcal{J}(\theta) = \frac{1}{kG} \sum_{j=1}^k M_j$
      backpropagate loss and take gradient step w.r.t. $\theta$
    **end for**
  **end while**
  **return** $\pi_\theta$

---

Low-discrepancy sampling induces the desirable property that in the limit $k \to m$, we fully recreate the original GRPO objective. In other words, one can achieve higher fidelity by scaling the amount of compute. We investigate this tradeoff further in Section 5.2.

## 5 EXPERIMENTS

To empirically validate our proposed algorithm, we start from the open source base model LLaDA-8B-Instruct (Nie et al., 2025) and fine tune models using AGRPO on three different reasoning tasks: GSM8K, MATH, and Countdown. GSM8K/MATH are standard problem-solving benchmarks consisting of 8.5k/12.5k math problems at the grade school/high school level, respectively (Cobbe et al., 2021; Hendrycks et al., 2021). Countdown is a popular math reasoning task where the model is given a list of 3-4 numbers and a target number; the goal is to combine the numbers using arithmetic operations $(+, -, \times, /)$ and parentheses to get the target number (Pan et al., 2025).

Due to compute/memory constraints, we use Low-Rank Adaptation (Hu et al., 2021) instead of full fine-tuning. We fix the response length at $n = 384$ and the number of steps at $m = 128$ for GSM8K/MATH and $m = 192$ for Countdown so that 2-3 tokens are unmasked per step. This balances inference wall time and stability, both of which are important for online RL. Crucially, despite being trained on a single configuration, we observe that model performance generalizes to different output lengths and steps.

During training, we generate rollouts with random remasking to inject stochasticity into the environment and incentivize exploration. Random remasking makes additional sense in the context of AGPRO since two rollouts with the same text can have different intermediate states at timestep $t$, helping the model learn from diverse contexts. We switch to low confidence remasking (which can be thought of as a form of annealing (Nie et al., 2025)) for evaluation. Other hyperparameters, including $G, \varepsilon$, and $\beta$, can be found in Appendix A.1.

### 5.1 RESULTS

**Strong reasoning improvements.** We report accuracies on test splits in Table 1. For diffusion models, AGRPO achieves the highest accuracy across all three tasks, comfortably beating the base LLaDA model and all other dLLM post-training methods, including PADRE (Shankar, 2025), VRPO (Zhu et al., 2025), and DCoLT (Huang et al., 2025); we refer readers to Appendix C for more detailed comparisons of these methods. These findings confirm our hypothesis that an RL method

Table 1: Test accuracies for RL post-training methods across different reasoning tasks and generation lengths. All tasks are 0-shot pass@1; the best accuracy for each model type and task is **bolded**. Outputs of length $n$ are generated with $m = n/2$ steps, except for entries marked with *, which use $m = n$ steps. Models trained on a larger mix of reasoning data are denoted with †, and models trained with LoRA (rather than full fine-tuning) are denoted with ‡. Missing entries (-) indicate that the corresponding paper did not report results for that task/length.

| Model (AR) | GSM8K | MATH |
|---|---|---|
| DeepSeekMath-Base 7B | 64.2 | 36.2 |
| + CoT SFT† | 82.9 | 46.8 |
| + CoT SFT + GRPO† | **88.2** | **51.7** |

| | GSM8K | | MATH | | Countdown | |
|---|---|---|---|---|---|---|
| Model (diffusion) | 256 | 512 | 256 | 512 | 256 | 512 |
| MMaDA (UniGRPO)† | - | 73.4* | - | 36.0* | - | - |
| LLaDA-8B-Instruct | 79.3 | 79.0 | 36.0 | 37.2 | 10.4 | 14.6 |
| + CoT SFT‡ | 78.8 | 81.1 | 32.6 | 34.8 | 14.5 | 23.8 |
| + diffu-GRPO‡ | 79.8 | 81.9 | 37.2 | 39.2 | 31.3 | 37.3 |
| + PADRE‡ | - | 85.6 | - | 40.9 | - | - |
| + VRPO† | 81.4 | 81.1 | 36.2 | 38.0 | 13.0 | 14.8 |
| + DCoLT‡ | 84.7 | - | **44.9***  | - | - | - |
| + AGRPO (ours)‡ | **87.3** | **89.0** | 42.2 | **43.8** | **40.0** | **41.3** |

built on unbiased policy gradients and exact likelihoods can learn stronger reasoning abilities than methods built on biased, unprincipled approximations.

Although performance on MATH still lags behind autoregressive models, we are able to achieve parity with DeepSeekMath-RL on GSM8K, a substantial improvement for dLLMs. Additionally, we observe increased performance gains from AGRPO (+10.0% on GSM8K and +6.6% on MATH) over GRPO (+5.3% on GSM8K and +4.9% on MATH), suggesting that increased training compute from Monte Carlo sampling leads to more effective learning and generalization.

**Inference tradeoffs.** A big advantage of dLLMs is their ability to trade off compute and quality at inference time. We examine how AGRPO affects the inference compute/quality frontier on GSM8K by fixing the response length at $n = 384$ and varying the number of sampling steps $m$. As shown in Figure 3, not only does AGRPO consistently achieve higher performance across all sampling steps, it matches the baseline with *6x fewer sampling steps*, a remarkable speedup. We give sample responses for different values of $m$ in Appendix B.

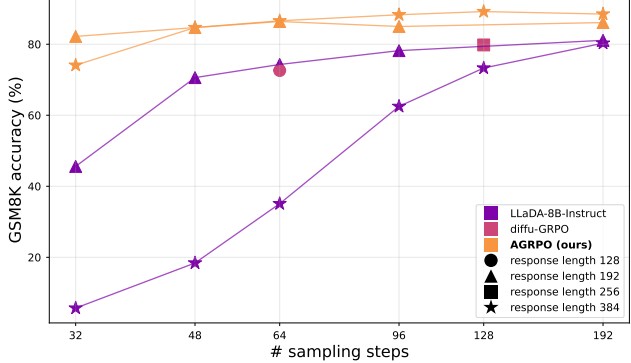

Figure 3: The inference compute/quality frontier for GSM8K across different configurations. Lines connect comparable points, i.e. same model and response length, showing the possible tradeoffs at inference time.

(a) Rewards for GSM8K over 600 steps. Note that the shaded area represents intra-run variance over a rolling window, not averaged across different seeds.

| $k$ | Wall time (s) per gradient step |
|----|----|
| 2 | 247.5 |
| 8 | 292.2 |
| 32 | 508.4 |

(b) Wall times averaged over all steps. Values are reported using a 8xA5000 GPU setup with a global batch size of 64 and $m = 128$ sampling steps.

Figure 4: Reward curve and wall time comparisons for different values of $k$ during a single GSM8K training run.

These results demonstrate not only that post-training with AGRPO can instill robust reasoning skills, but also that these reasoning skills can complement dLLMs' innate ability to trade off inference compute and quality. To our knowledge, this is the first investigation of how post-training dLLMs affects inference tradeoffs by expanding the Pareto frontier, opening up a new perspective on the benefits of post-training.

## 5.2 ABLATIONS ON K

When choosing the number of Monte Carlo samples $k$ to use, the immediate tradeoff is clear: increasing $k$ lowers variance but costs more compute. Another important (but perhaps less obvious) point to consider is the inference cost: in online RL, generating rollouts is often more expensive than computing the actual policy update ($m$ forward passes per rollout vs. 1 forward/backward pass per loss computation). This effect is magnified for dLLMs, since full self-attention and lack of KV caching means that inference is especially compute-heavy. Thus we can choose moderately large $k$ without significantly increasing compute, as long as $k \ll m$.

Empirically, we do indeed observe faster convergence for larger values of $k$ at the cost of increased wall time, as seen in Figure 4. Note that due to the constant inference costs mentioned above, the compute/performance tradeoff induced by $k$ is highly nonlinear. As a result, although smaller values of $k$ spend up to 2x less compute per step, it takes far more than 2x as many steps to reach the same reward level. Pinpointing the exact relationship between $k$ and downstream performance (e.g. final test accuracy on FLOP-matched runs) is an area we identify for future work.

## 6 CONCLUSION

This work presents AGRPO, an online RL algorithm designed for dLLMs and grounded in the multi-step diffusion perspective of RL. AGRPO computes unbiased policy gradient estimates via Monte Carlo sampling, making it the first policy gradient method that is both principled and tractable for dLLMs. Using our proposed algorithm, we show how to effectively train dLLMs, beating comparable methods across multiple tasks and redefining the inference compute/quality frontier. These contributions establish AGRPO as a viable way to transfer policy gradient RL techniques to the dLLM setting; we hope future works can build on our methods, either theoretically or empirically, and further close the gap between dLLM and AR LLM post-training.

## REPRODUCIBILITY STATEMENT

We present pseudocode and a discussion of implementation details in Algorithm 1 and Appendix E, respectively. The high-level setup for experiments is given in Section 5. We refer readers to

the appendix for a more detailed report of datasets, hyperparameters, implementation, and sample responses.

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

# A EXPERIMENT DETAILS

## A.1 TRAINING

See Table 2 for a full list of training hyperparameters.

Models are trained on 8xA5000 GPUs for 800-1200 steps (depending on the task). We use Hugging Face's TRL framework (von Werra et al., 2020) to implement the training code. Rewards are a

Table 2: Comprehensive list of hyperparameters used during training.

| Parameter | GSM8K | MATH | Countdown |
|---|---|---|---|
| Response length $n$ | 384 | 384 | 384 |
| # sampling steps $m$ | 128 | 128 | 192 |
| MC samples $k$ | 32 | 24 | 24 |
| Responses per group $G$ | 8 | 16 | 16 |
| # gradient iterations $\mu$ | 1 | 1 | 1 |
| Clipping threshold $\varepsilon$ | 0.2 | 0.2 | 0.2 |
| KL coefficient $\beta$ | 0.01 | 0.01 | 0.01 |
| Temperature | 0.5 | 0.5 | 0.6 |
| LoRA $r$ $(= \alpha)$ | 8 | 8 | 32 |
| Batch size | 64 | 64 | 64 |

combination of rule-based formatting rewards, i.e. the presence of `<think>` and `<answer>` tags, as well as ground-truth verified rewards.

## A.2 EVALUATION

For each task, we use the last checkpoint for evaluation. Surprisingly, we found that models trained on GSM8K problems achieved higher accuracy on MATH500 test problems than models trained on the MATH training split. The accuracy reported in Table 1 is thus from the GSM8K model; models trained on MATH showed more modest gains, achieving 40.8% at $n = 512$. We hypothesize that the more difficult nature of MATH, combined with inherent limitations of the pretrained LLaDA model, make it harder to learn generalizable reasoning abilities during RL.

We also note that baseline numbers for LLaDA-8B-Instruct differ slightly from Zhao et al. (2025) and Nie et al. (2025), most likely due to differences in prompt, batch size, etc., although this doesn't substantially change the results presented.

## A.3 DATASETS

For all tasks and train/test splits, we use open source datasets on HuggingFace. Accuracies are reported on the GSM8K-Platinum (Vendrow et al., 2025) test split, a cleaned version of the original GSM8K test split, and the MATH500 subset of the overall MATH benchmark. We train on the first 10000 problems of the Countdown dataset[4] and save the last 1000 for evaluation.

We use HuggingFace's math-verify library for parsing GSM8K and MATH answers.

## B SAMPLE RESPONSES

See Table 3 for the system prompt as well as sample responses from a random GSM8K test problem. As evident in the responses, RL induces a rigid step-by-step structure, which we hypothesize allows the model to arrive at the same solution with many less steps (although there are some clear artifacts in the $m = 48$ solution, such as the repeated zeros).

## C COMPARISONS TO OTHER DLLM POST-TRAINING METHODS

Here we continue our discussion of previous dLLM post-training methods presented in Section 5.

**PADRE.** Shankar (2025) introduces pseudo-likelihood based alignment for dLLMs, where the pseudo-likelihood of a sequence $x$ is defined as the product of conditional marginals $\prod_i p(x_i \mid x_{-i})$ where $x_{-i}$ replaces the $i$-th token of $x$ with ■. This likelihood is used to minimize KL divergence

---

[4] https://huggingface.co/datasets/Jiayi-Pan/Countdown-Tasks-3to4

Table 3: Sample responses ($n = 384$) for different values of $m$ (# sampling steps) from a model trained with AGRPO. Prompt is from the GSM8K-Platinum dataset.

| |
|---|
| **System:** You will be given a math problem. Think through the problem step by step, enclosing your reasoning process in <think>tags. Provide your final answer in <answer>tags, with only the number or expression enclosed in \boxed{}, as follows: <think>... </think><answer>\boxed{...} </answer> |
| **User:** 44 seniors need to receive awards. Each senior receives a picture frame that costs \$20. Each picture frame needs to be etched with the logo for an additional 20% cost per frame. 2 of the seniors will also receive pins that are \$5. 1/4 of the seniors are officers and they will need to receive cords that are \$12 each. How much will be spent on the senior gifts? |
| **Assistant (m=192):** <think>1. First, calculate the cost of each etched picture frame: - The cost of one picture frame is \$20. - The additional cost for etching is 20% of \$20, which is $0.20 \times 20 = 4$. - Therefore, the cost of each etched picture frame is $20 + 4 = 24$. 2. Calculate the total cost for the etched picture frames: - There are 44 seniors, and each receives an etched picture frame costing \$24. - The total cost for the picture frames is $44 \times 24 = 1056$. 3. Calculate the total cost for the pins: - 2 seniors will receive pins, and each pin costs \$5. - The total cost for the pins is $2 \times 5 = 10$. 4. Calculate the number of officers: - 1/4 of the seniors are officers, so the number of officers is $frac14 \times 44 = 11$. 5. Calculate the total cost for the cords: - Each cord costs \$12. - The total cost for the cords is $11 \times 12 = 132$. 6. Calculate the total amount spent on the senior gifts: - Add the total costs for the picture frames, pins, and cords. - The total amount spent is $1056 + 10 + 132 = 1198$. </thinkthink><answer>\boxed{1198} </answer> |
| **Assistant (m=48):** <think>1. First, calculate the cost of the picture frames: - Each base picture frame picture costs \$ \$20 
 - Etching costs an additional 20% of \$ so0, 0.2000000000000000 20 = 4. Therefore, the total cost per picture frame is 20 + 4 = 24. 
 - Total cost for 44 seniors is 44 * 24 = 1056. 2. Next, calculate the cost of pins: - 2 seniors will receive pins, and each pin costs \$5. - Total cost for pins is 2 * 5 = \$10. 3. Finally, calculate the cost of cords: - 1/4 of the seniors are officers, so there are 44 / 4 = 11 officers. - Each cord costs \$12. - Total cost for cords is 11 * 12 = \$132. 4. Add, up all the costs: - Total cost = picture frames is \$1056 + pins is \$10 + cords is \$132. - Total cost = 1056 + 10 + 132 = \$1198. </think><answer>\boxed{1198} </answer> |

between the current policy and a reward-optimal policy $\propto \pi_{\text{ref}}(x) \exp(r(x)/\beta)$, exploiting the fact that two distributions are equal iff their pseudo-likelihoods are equal. The model effectively only learns from a single-masked state, whereas AGRPO allows the model to learn from all intermediate masking states.

**VRPO.** Introduced by Zhu et al. (2025) as part of the training process for LLaDA 1.5, Variance-Reduced Policy Optimization follows a similar approach to Direct Preference Optimization, where the model directly learns from pairwise preference data without external rewards. The VRPO objective consists of the DPO objective with all likelihoods replaced by ELBOs, along with several variance reduction techniques. Overall, the LLaDA 1.5 paper studies a slightly different setting (alignment/RLHF instead of reasoning/RLVR) and also relies heavily on likelihood bounds.

**DCoLT.** Inspired by autoregressive chain-of-thought methods, Huang et al. (2025) optimize a Diffusion Chain of Lateral Thought. Instead of considering sequence-level actions, they adopt a multi-step denoising view where an action consists of unmasking $n/m$ tokens (similar to our approach, which could explain the similar performance). However, there are several key differences that make AGRPO training much simpler than DCoLT: there is no need to train a separate module that ranks

which token positions to unmask, and we can *sample* timesteps to estimate gradients rather than accumulating gradients over all $m$ timesteps.

# D  LOW-DISCREPANCY SAMPLING DETAILS

An outline of Zheng et al. (2025)'s discrete low-discrepancy sampler is given below.

1. Sample $u_j$ i.i.d. from $U([0,1])$ for $j = 0$ to $k - 1$.
2. "Bin" them into $k$ disjoint bins by defining $u'_j = (u_j + j)/k$.
3. Define the final samples $t_j \in \{1, \ldots, m\}$ as $t_j = \lfloor mu'_j \rfloor + 1$.

This sampler allows us to "cover" a wide range of timesteps from $0$ to $m$ when estimating the AGRPO objective. We can also extend this to the batch level: instead of sampling $k$ timesteps, we sample $bk$ timesteps where $b$ is the batch size.

# E  PRACTICAL CONSIDERATIONS

In this section, we discuss in detail various decisions and tradeoffs made regarding the actual implementation of AGRPO. As with all online RL algorithms, the goal of any implementation is to run as efficiently as possible while maximizing efficacy.

**Caching partially masked states.** In order to obtain exact action/token likelihoods, we must recreate the exact state/context at the step where that token was unmasked. To do this efficiently, we cache the unmasking order during generation so that each token is associated with a timestep $t$. Then, to get the partially masked state at timestep $t$, we simply mask out all tokens with timestep $\geq t$.

**Memory-efficient gradient accumulation.** Naively computing the AGRPO objective (equation 4) and backpropagating would require keeping $k$ forward passes simultaneously in memory, which can quickly saturate GPU HBM for large models. Instead, one can accumulate the gradient immediately after each MC sample is computed by calling `loss.backward()` (without taking an optimizer step) *inside* the for loop. This frees the computational graph and avoids excess memory usage.

**float64 Gumbel-based categorical sampling.** dLLMs typically use the Gumbel-max trick to sample from output logits. However, Zheng et al. (2025) point out that naively using `float32` causes an inconsistency between theoretical and actual behavior due to numerical instability. We follow their recommendation of using `float64` for the sampling stage.

**Handling EOS tokens.** Since dLLMs generate with a fixed number of sampling steps, in the later stages, the model can spend many "garbage" steps producing `EOS` tokens at the end of a sequence (while other sequences in the batch are still generating useful tokens). Gradient updates on these steps don't provide meaningful information to the model, so we set the max timestep in our low-discrepancy sampler (Section D) to be the last timestep a non-`EOS` token was generated.

## E.1  LIMITATIONS

Since AGRPO involves multiple forward passes per gradient step, it is inherently quite compute-hungry, and training can be quite slow in terms of wall time. Furthermore, there are many potential ways to estimate the GRPO objective with Monte Carlo sampling; sampling along the time axis is a natural way, but there could other ways, e.g. importance sampling, that optimally reduce variance.

Empirically, we observe that some models trained with AGRPO that achieve high reward can exhibit signs of entropy collapse, namely reduced token diversity and formulaic, "template"-y answers.

# F  REMARKS ON DLLM INFERENCE

Despite a more complicated training setup, dLLMs enjoy several potential benefits at inference time: they can generate text in arbitrary order, are naturally self-speculative (i.e. one can see the model's best guess for the entire sequence at every step), and can trade off compute and generation quality by choosing to unmask more or less tokens per step.

Here we discuss several unique characteristics of dLLM inference, which may be helpful to readers who are only familiar with traditional AR LLM inference.

**Fixed sequence length.** One drawback of dLLMs is that the context length must be fixed ahead of time, instead of being dynamically grown as with AR LLMs. Works such as BlockDiff address this issue by introducing a hybrid autoregressive/diffusion framework (Arriola et al., 2025); in this paper, we stay within the normal diffusion framework for simplicity.

**Instruct-tuned models.** dLLMs such as LLaDA-8B-Instruct, which have undergone supervised fine-tuning (SFT) on instruction-following traces, tend to place higher probabilities on EOS tokens. When combined with low-confidence unmasking, this leads to an unnaturally high proportion of EOS tokens in later positions and terse, stilted responses (Nie et al., 2025). Thus, to generate text with standard left-to-right prompting, we divide the response into smaller blocks (e.g. 32 tokens), unmask tokens within the leftmost block, and continue to the next block once all tokens in the current block have been unmasked. This is known as *semi-autoregressive* sampling (Yang et al., 2025).

**Bidirectional prompting.** In this paper, we work with traditional left-to-right prompting, which is the native format for the reasoning datasets we use. This leaves a big dLLM advantage on the table — namely their ability to generate text from arbitrary context. Future works could consider reasoning tasks that involve using context from both the left and right; for example, giving the model a problem and the numerical answer, and forcing it to deduce the intermediate steps.

**KV caching.** Since diffusion transformers typically use full self-attention instead of causal self-attention (Peebles & Xie, 2022), embeddings for the same token position can change depending on the sampling step. This non-causality prevents dLLMs from using the same KV caching mechanism as AR LLMs. As a result, generating same-quality text with dLLMs is significantly slower than same-scale AR models, which is especially painful for online RL. However, there has been some recent interest in KV caching alternatives for dLLMs (Wu et al., 2025; Ma et al., 2025). Since dLLMs already have the ability to decode multiple tokens in parallel, we believe a successful implementation of KV caching is imperative to realizing dLLMs' potential as a faster, more flexible alternative to AR models.

