# OpenReview forum: "Principled and Tractable RL for Reasoning with Diffusion Language Models"
_ICLR.cc/2026/Conference — Submitted to ICLR 2026_

### Official Review · Reviewer_9uBq · 2025-11-01

**Soundness:** 2
**Presentation:** 1
**Contribution:** 1
**Rating:** 2
**Confidence:** 3

**Summary:**

The paper studies the problem of on-policy reinforcement learning for diffusion language models. In dLLMs, computing marginal token probabilities is intractable, making autoregressive LLM RL objectives not directly applicable. The authors point out the approximations and bias in existing diffu-GRPO and UniGRPO objectives. To tackle the issues, the authors proposes Amortized GRPO (AGRPO), which rewrites GRPO’s tokenwise inner sum as an expectation over timesteps and estimates it with Monte-Carlo samples, resulting in an unbiased policy-gradient estimator. The objective also keeps the KL term inside the expectation approximated with the Schulman estimator. The method is paired with practical tricks (caching partially masked states, LoRA) for efficiency. AGRPO is tested with LLaDA-8B-Instruct across GSM8K, MATH, and Countdown, improving over the base model and outperforming diffu-GRPO and UniGRPO.

**Strengths:**

* The paper outlines the problem fairly clearly, outlining issues with existing methods and proposing a relatively straightforward algorithms to address them.
* The authors also provide some details about the practical considerations which are useful.
* While I have various concerns over the empirical setup detailed below, the results seem promising.

**Weaknesses:**

* The paper misses some closely related prior work on RL fine-tuning of diffusion language models [1, 2, 3]. I believe a comparison to these baselines would be critical. (There are several other recent papers studying the same problem but they count as concurrent work, so I do not expect the authors to compare with them)
* There are several explanations and references in the paper that are incorrect:
    * L184: For early work on RL with LLMs the authors refer to the WizardLM paper as a reference for the use of PPO. However, there were quite a few papers [e.g. 4] prior to that.
    * Initial work on RL with LLMs was also focused on preference learning rather than reasoning ability in math or code.
    * L190: Learning reward models is still done for preference tasks and it is usually decoupled from the size of the model being trained so it is not as affected by the growing model sizes.
* In terms of baselines, simple test-time scaling methods like majority voting on the base-model are missing.
*  A lot of details around the actual runtime (e.g. how many steps of training, comparison of training time vs baselines) are missing. Additionally it seems that the numbers in Table 1 are the result of a single seed of training and single seed for evaluation. Please clarify if this is incorrect. I understand training with multiple seeds can be expensive but I atleast expect multiple seeds for eval.
*  There are no ablations or analysis to investigate factors affecting how the method works. For instance, analysis on the effect of k, m, n (during training), effect of low confidence remasking.
*  Finally, the results are limited to a single model with 3 math-based tasks which is hard to get general insights from.

[1] Venkatraman et al., 2024. Amortizing intractable inference in diffusion models for vision, language, and control.

[2] Huang et al. 2025. Reinforcing the Diffusion Chain of Lateral Thought with Diffusion Language Models.

[3] Zekri and Boullé, 2025. Fine-Tuning Discrete Diffusion Models with Policy Gradient Methods.

[4] Ziegler et al., 2019. Fine-Tuning Language Models from Human Preferences.

**Questions:**

* Please clarify if the numbers are averaged over multiple seeds.
* In Fig 3, why are the numbers for the base model for different response lengths missing?

---

> ### Author Response · Authors · 2025-11-28
> **Response to Reviewer 9uBq**
>
> We sincerely thank the reviewer for their careful analysis and thoughtful feedback on our paper. We hope to address some of the reviewer's concerns below.
>
> ### Weakness 1: comparison to recent works
> - The manuscript has been updated to include more comparisons to recent dLLM post-training works. Specifically, we include PADRE, VRPO, and DCoLT (a suggested inclusion). We hope these comparisons provide helpful insight on how AGRPO allows for powerful, robust RL training.
> - Venkatraman et al. (2024) focus on the problem of posterior sampling given a diffusion prior, which is a much more general setting than RLVR post-training of dLLMs. They report results on image generation, text infilling, and robotics, which are quite different in nature to mathematical reasoning tasks, making direct comparison hard.
>     - We do note that Appendix H.1 in their paper alludes to the same timestep-sampling trick that we use in AGRPO.
> - Zekri and Boullé (2025) build on the score entropy/CTMC formulation of discrete diffusion. This is a more general/complex formulation than the masked diffusion formulation that is now generally accepted for language (e.g. LLaDA). Since AGRPO, diffu-GRPO, and other comparable methods all assume things like masked transitions and discrete time, which are specific to LLaDA, we believe the other works included in the paper provide a more illuminating comparison.
>
> ### Weakness 2: references to early RL LLM works
> - We thank the reviewer for these comments; we have revamped Section 3.2 accordingly to better showcase the transition from early RLHF works to more recent RLVR works. A citation of Ziegler et al. (2019) has been added.
>
> ### Weakness 3: lack of training/evaluation details
> - We have added more details (e.g. # of training steps) in Appendix A.1. Note that numbers for other dLLM post-training methods are taken from the respective paper for simplicity, and their training runs may have different hardware, hyperparameters, etc., so directly comparing training time is not straightforward.
> - Numbers in Table 1 are from a single training seed. Following other works, models are evaluated using greedy decoding (e.g. temperature=0), so evaluation is deterministic, i.e. there is no need to average across multiple eval seeds. Furthermore, most works on RLVR with dLLMs (d1, PADRE, DCoLT, MMaDA) omit baselines such as pass@k or maj@k, as the focus is mainly on improving the RL algorithm itself.
>
> ### Weakness 4: limited tasks/ablations
> - Although the number of tasks is less than those commonly found in the AR post-training literature, we believe using 3-4 tasks is standard practice for dLLMs, including d1 (our main baseline) and almost every other related work we include, so this number doesn't hurt comparison. The limited tasks mostly reflect the developing nature of dLLM post-training compared to AR post-training rather than a lack of rigor.
> - The updated manuscript includes Section 5.3, which studies how ablating k (the central quantity in AGRPO) affects training compute and performance.
>
> Question: **In Fig 3, why are the numbers for the base model for different response lengths missing?**
>
> We have updated Figure 3 to include 2 response lengths for both AGRPO as well as the base LLaDA model. We also note that the d1 paper only reports results for a single # of sampling steps and there is no open checkpoint available, which prohibits our paper from showing the entire inference frontier for diffu-GRPO.

---

### Official Review · Reviewer_PuSn · 2025-11-01

**Soundness:** 3
**Presentation:** 2
**Contribution:** 2
**Rating:** 4
**Confidence:** 3

**Summary:**

This work presents Amortized Group Relative Policy Optimization (AGRPO), a novel on-policy reinforcement learning algorithm tailored to advance the reasoning abilities of Diffusion Large Language Models (dLLMs).  This paper identify a key limitation in current RL approaches: while proven effective for autoregressive LLMs, they become computationally prohibitive for dLLMs, and existing dLLM-specific methods often resort to heuristic approximations that introduce bias into policy gradients and lack theoretical grounding. To address this, AGRPO reformulates the token-wise summation in the policy gradient objective as an expectation, enabling unbiased and efficient estimation through Monte Carlo sampling over generation timesteps. Empirical evaluations on mathematical reasoning tasks, including GSM8K, show that AGRPO substantially outperforms both baseline models and prior RL techniques, offering a theoretically rigorous framework that not only elevates dLLM performance but also rebalances the interplay between computational efficiency and output quality.

**Strengths:**

1. The experimental results are impressive, achieving high performance on the GSM8K benchmark (86.3) and matching the performance of models such as DeepSeekMath-Base 7B.
2. The core idea is novel and well-motivated, with several practical considerations incorporated into the methodology. I found the approach insightful and instructive.

**Weaknesses:**

1. While the idea is interesting, the writing requires improvement in several areas. Descriptions in many parts remain unclear, and the experimental section appears somewhat brief. The paper would benefit from more comprehensive ablation studies—for instance, to validate the impact of the practical considerations mentioned. A restructuring of the paper is also important to better align with academic writing standards.
2. The version of paper currently lacks a dedicated "Related Work" section, which is essential for contextualizing the contribution within the existing literature.
3. Related to the first point, the current version reads more like a technical report than a fully developed academic paper.

**Questions:**

1. Can this method be effectively adapted to other diffusion-based language models, such as Dream 7B?

---

> ### Author Response · Authors · 2025-11-28
> **Response to Reviewer PuSn**
>
> We thank the reviewer for their thoughtul comments and for recognizing the core ideas of AGRPO as novel, well-motivated, and empirically promising. We would like to address some of the concerns presented here:
>
> ### Weakness 1: writing
> - The manuscript has been newly updated (see official comment) with improved writing in many sections, including Sections 1 and 3. In particular, more detailed explanations have been given for readers potentially unfamiliar with the theoretical assumptions of diffusion.
> - We have also added an ablation on how k, the number of MC samples, affects training compute and performance.
>
> ### Weakness 2: lack of related works
> - Although there is no section in the paper titled "Related works", we believe readers will gain a clear sense of where the field stands on diffusion language models and RL for post-training through Sections 2, 3 and Appendix C. These sections describe the foundations of dLLM training/inference and the historical context of RLVR for reasoning as well as several related methods for post-training dLLMs. With these sections, we believe there is no need to separately dedicate a section for related works.
>
> Question: **Can this method be effectively adapted to other diffusion-based language models, such as Dream 7B?**
>
> Yes, AGRPO can be extended to any masked diffusion model, including LLaDA and Dream 7B. We chose LLaDA as it was the most widely known open-source dLLM at the time of writing, but AGRPO is theoretically compatible with any pretrained diffusion language model.

---

### Official Review · Reviewer_2DXg · 2025-11-03

**Soundness:** 2
**Presentation:** 2
**Contribution:** 2
**Rating:** 2
**Confidence:** 4

**Summary:**

This paper presents AGRPO (Amortized Group Relative Policy Optimization), a rl algorithm designed to post-train diffusion language models on reasoning tasks by reformulating the GRPO objective as an expectation over denosing timesteps that can be estimated via Monte Carlo sampling. Empirically, AGRPO has accuracy gains on math/logic benchmarks over prior policy optimization methods for dLLMs, trading extra compute for per-timestep estimates in a way that aims to reduce bias/variance in log prob estimation.

**Strengths:**

1. this paper discusses caching partially masked states, gradient accumulation per MC sample, and EOS/timestep handling, which are useful for training diffusion LLMs.
2. this paper shows their proposed method achieves better performance than baselinse UniGRPO and diffu-GRPO across 3 tasks.
3. This paper trades off compute (number of MC estiimates) with approximation accuracy and achieves better results than fewer MC sample baselines.

**Weaknesses:**

The paper’s writing is not rigorous:

1. Equation 2 is AR-specific and assumes a causal left-to-right generation; mapping its per-token inner sum to dLLM timesteps needs clearer justification. As written, the swap (|o_i|\to m) can mislead readers about where compute actually occurs in dLLMs.

2. Exact sequence likelihood in dLLMs is intractable because it would require marginalizing over all denoising orders/masking patterns, but AGRPO doesn’t do that: it optimizes using conditionals on partially-masked states and estimates an ELBO-based surrogate via MC. MC is unbiased for the ELBO, but the ELBO itself is a biased lower bound to the true log-likelihood. The paper should avoid calling such MC "exact". Additionally, optimizing the particular sampling order might not be optimal, because if a response is good, we ideally want to increase the probability of generating this response with all possible orderings to generalize better. UniGRPO and diffu-GRPO resample new masking patterns (randomly or full mask) can somehow encourage this.

3. The notation $$(\pi_\theta(o_t \mid q, o_{<t}))$$ overloads AR conventions; in dLLMs (o_{<t}) should be defined as the partially masked state at timestep t. The paper uses this shorthand broadly without a crisp definition, which hurts clarity.

4. this paper also lacks a related work section

Limited novelty and lack of experiments:

1. AGRPO replaces one-step heuristics with an unbiased per-timestep estimator—a principled step—but the paper lacks ablations on k (MC samples), estimator variance/bias, and how k drives compute–quality trade-offs. Without these, the incremental benefit over prior heuristics is harder to quantify. The experiments are lacking rigorous ablations and for the main results, the tasks they choose are a subset of prior works.

2. Increasing the number of MC samples introduces additional compute, the paper doesn't analyze the efficiency and performance gain trade off.

**Questions:**

see above.

---

> ### Author Response · Authors · 2025-11-28
> **Response to Reviewer 2DXg**
>
> We sincerely thank the reviewer's thorough feedback and recognition of AGRPO's empirical strength over other methods that rely on one-step approximations. We address some of the concerns here:
>
> ### Weakness 1: confusing notation
> - In the updated manuscript, the notation has been changed to better reflect the diffusion setting. Specifically, the $o\_{<t}$ notation has been dropped since it can confuse readers who are only familiar with the AR setting. We instead use $o_t$, which is consistent with conventions in the diffusion literature for an intermediately noised state at time $t$.
> - Verbal descriptions in Sections 3.3 and 4.1 have been improved to clearly explain key mathematical expressions. We clarify that $|o_i|$ refers to the number of sampling steps, which can vary from response to response for AR models but is fixed for diffusion, hence why we can assume $|o_i|=m$ for dLLMs.
>
> ### Weakness 2: inexact likelihoods
> - We point out that this is interprentation of our approach is **incorrect** and most likely stems from a misunderstanding of the underlying setup, aka a multi-step denoising MDP. We have edited Section 3.1 to more explicitly frame the setup; to reiterate, we treat the diffusion model as a policy that takes an action by performing a single unmasking step.
> - Because diffusion models parameterize conditional marginal distributions of intermediate masked states, we have exact likelihoods, so there is no need for an ELBO. Thus the MC estimator does not estimate a lower bound, but rather an unbiased version of the policy gradient objective.
>    - Interestingly, although the reviewer initially claims the MC estimator is biased, they later note that "AGRPO replaces one-step heuristics with an unbiased per-timestep estimator—a principled step," suggesting there is some confusion about the actual theoretical content of the paper.
> - Although the unmasking order is fixed for each gradient step, low-discrepancy sampling (Section 4.2) still guarantees that the model learns to unmask from a variety of timesteps/masking levels. Ultimately, even though diffu-GRPO and UniGRPO can resample masks, AGRPO achieves superior performance, suggesting that the fixed unmasking order is not a limitation.
>
> ### Weakness 3: lack of novelty
> - The new manuscript (see official comment) better emphasizes the novel contributions of AGRPO, including a different way of framing the underlying MDP to align with the denoising process, a principled way of estimating the policy gradient objective, and a new way of investigating how post-training affects dLLMs' ability to trade off inference compute and quality.
> - A new section (5.3) has been added to better describe how increasing $k$ affects training dynamics and compute/wall time. This section should help support AGRPO's ability to trade off efficiency and performance.
> - We note that using 3-4 reasoning tasks is standard in the nascent dLLM post-training literature, including d1 (our main baseline) and almost every other related work we include. We believe this number mostly reflects the developing nature of dLLM post-training compared to AR models, rather than a lack of rigor.
>
> ### Weakness 4: no related works
> - Although there is no section formally titled "Related works", we believe readers who fully read and understand Sections 2, 3 and Appendix C will gain a clear sense of where the field stands on diffusion language models and RL for post-training. These sections describe important works that lay the foundations for dLLM training/inference and trace the historical background of RLVR for reasoning; thus, there is no need to separately dedicate a traditional "Related works" section.

---

### Official Review · Reviewer_3eva · 2025-11-06

**Soundness:** 1
**Presentation:** 2
**Contribution:** 1
**Rating:** 0
**Confidence:** 4

**Summary:**

The paper identifies a key issue in post-training diffusion large language models (dLLMs), that the existing RL methods like GRPO, which work well for autoregressive (AR) models, are incompatible with dLLMs because dLLMs cannot compute probabilities in a single forward pass. It introduces Amortized Group Relative Policy Optimization (AGRPO), a reinforcement learning (RL) algorithm designed for fixing this, by reformulating the GRPO objective by treating the sum over timesteps as an expectation, which is then estimated using Monte Carlo sampling over timesteps. They claim this provides an unbiased policy gradient. The paper demonstrates performance gains on mathematical reasoning tasks (GSM8K, MATH, Countdown) over prior dLLM RL methods.

**Strengths:**

The core idea of trying to reformulate GRPO  to enable Monte Carlo estimation is new. If correct,  it can directly addresses the basic incompatibility between standard policy gradient methods and the dLLM architecture.
The paper is generally well-written and clearly explains the theoretical shortcomings of existing approaches. There are additional tricks describe for efficient computation which can be valuable in other contexts as well.

**Weaknesses:**

I might have misunderstood the key idea — but it seems to be based on a wrong interpretation of what the actual incompatibility between PGRL methods and dLLMs. They use the AR-incorporated PPO/GRPO loss where the propensity ratio has been simplified because of autoregressive per token likelihood decomposition. This itself becomes invalid for dLLMs. While some form of regressive factorization is still true, the model likelihood will still not match the true likelihood. This makes AGRPO another approximation (which is fine, but the paper does not claim that). Furthermore even if this term is unbiased (which I do not believe to be the case), the KL term still requires true log-likelihoods, which is not what the AGRPO  trick allows to compute. Overall the description sidesteps the fundamental issue that dLLMs do not have a tractable sequence-level probability.

Additionally, the results compare only a specific length case. While compute resource constraints make sense for training, inference is done with the same model for different lengths. So that does not seem to a big issue. Additionally they use LoRA based fine-tuning instead of full model tuning, which makes the comparison a bit apples-to-oranges. The gains can be from LoRA, and then the baselines need to be tuned with LoRA. Moreover the paper does not compare against more recent baselines such as [1,2,3], which also attempt to deal with the likelihood mismatch issue.

[1] wd1: Weighted Policy Optimization for Reasoning in Diffusion Language Models

[2] PADRE: Pseudo-Likelihood based Alignment of Diffusion Language Models

[3] DiffuCoder: Understanding and Improving Masked Diffusion Models for Code Generation

**Questions:**

Experiments:
I do not see the contribution of low-discrepancy sampling being ablated. How much does it actually help over naive i.i.d. sampling?
Similarly, usage of sampling t, produces variance, there should be an ablation of performance against t/m .


The paper claims the new loss is more efficient scalable, but I do not see the compute-cost vs gain tradeoff. Using the m-sample likelihood is more compute expensive then the 1-step approximation used in diffu-GRPO/d1's likelihood approximation. The cost/tractability is not just from samples but from overall FLOPS, which is not compared.

Relatedly, the argument that rollout generation cost "dwarfs" the loss computation is highly dependent and needs some empirical measures

Theory:
The paper uses the diffusion time t, as also a generated sequence length measure (by conditioning on $$o^i_{<t}$$ . How do you then generate that with independently sampled time $t \in [1,m]$. This is not an issue in AR models, as they generate tokens sequentially for t=1 to t=m; but that is not the case here. Is this multiple one-step approximations? Or something different?
What sequence/step likelihood is used for the KL computation in the objective.
Furthermore, the paper cites a blog-post for an unbiased KL estimate, but that post itself has multiple unbiased estimates. I think I know the version used, but this is not clear from the paper itself.
Since the code is not with the paper, I also cannot compare the expression used compared to the one in the paper to see if I misunderstood the expression; and the paper does not give enough details to know exactly what the computations are ( for both the importance ratio and KL likelihood)

---

> ### Author Response · Authors · 2025-11-16
>
> We thank the reviewer for their thoughtful feedback and suggestions. However, it seems there are some misunderstandings on the theoretical framing, which we hope to address here.
>
> It is true that dLLMs lack a tractable sequence-level likelihood which prohibits one from naively using the AR-factorized GRPO loss. However, we are crucially treating the RL problem as a **multi-step MDP** where the action space is **individual denoising steps**, which allows us to sidestep approximations and use exact action likelihoods. In other words, the term $\\pi_\theta(\cdot\mid q,o^i_{<t})$ is the exact policy distribution at intermediate step $t$.
>
> This framing has been used in prior works for diffusion RL; for example, see section 4.3 in [1]. Crucially, this is a different framing than the standard one-step framing, where the policy directly generates a distribution over final sequences, which is AR-friendly but not diffusion-friendly (as the reviewer notes).
>
> The KL regularization follows the same principle: by treating $\\pi_{ref}$ and $\\pi_\theta$ as distributions over intermediately-noised outputs, we get an unbiased estimator that we can sample over timesteps. Following the original GRPO paper [2], we use the KL estimator $D_{KL}(p||q)=\frac{q(x)}{p(x)}-\log\frac{q(x)}{p(x)}-1$.
>
> We plan to update the manuscript shortly, specifically the abstract and Section 3.1, to better emphasize this distinction between single- and multi-step MDPs, as this is a core assumption of AGRPO.
>
> [1] Kevin Black, Michael Janner, Yilun Du, Ilya Kostrikov, and Sergey Levine. Training diffusion models with reinforcement learning.
> [2] Zhihong Shao, Peiyi Wang, Qihao Zhu, Runxin Xu, Junxiao Song, Xiao Bi, Haowei Zhang, Mingchuan Zhang, Y. K. Li, Y. Wu, and Daya Guo. Deepseekmath: Pushing the limits of mathematical reasoning in open language models, 2024

---

> ### Author Response · Authors · 2025-11-28
> **Response to Reviewer 3eva**
>
> We thank the reviewer for their rigorous feedback. We are glad the reviewer found our approach novel, well-written, and practically useful. To address some of the concerns more concretely:
>
> ### Weakness 1: lack of tractable sequence likelihood
> - It is true that diffusion models lack an AR-like decomposition of the sequence likelihood. However, as noted in previous comments, we argue that this is not a critical flaw given our theoretical framing: in a multi-step denoising MDP, the action space is a single unmasking step rather than the entire sequence. Since diffusion models parameterize the conditional marginals of a partially masked sequence, we get **exact** action likelihoods at the cost of many more model evaluations. Our proposed MC sampling algorithm deals with this by sampling over timesteps.
>     - For AR models, it is common to view the problem as a one-step MDP where the policy generates the output in a single action, since we get sequence likelihoods for free. We argue that this is the incorrect approach for diffusion models, including prior works that try to approximate or bound the sequence likelihood.
> - This multi-step view also makes the KL penalty tractable, since we can compare on-policy and reference distributions from the same partially masked state.
>
> ### Weakness 2: limited experimental scope (single length, LoRA)
> - We have recently updated our manuscript to compare results across sequence lengths of **256 and 512**; see Table 1 for updated numbers.
>    - We also note that in the dLLM literature, it is not uncommon to report benchmark results for a single output length, for example in the LLaDA and MMaDA papers, as well the reviewer's suggested work PADRE.
> - It is true that exact performance numbers from LoRA can differ from full fine-tuning. However, we view this as a lower-level implementation detail that doesn't undermine our results; the increased reasoning abilities demonstrated by AGRPO over the base model are not LoRA-specific. Moreover, works such as d1 (which is our primary baseline) **also use LoRA**; we have added markings to Table 1 to better guide these comparisons.
> - We have added an ablation study on k, giving insight into how higher values of k affect the compute/performance tradeoff. These numbers also justify our claim that dLLM inference costs play a major role in RL compute, so that double k does not in fact double wall time.
>
> ### Weakness 3: comparisons to recent baselines
> - We have **added PADRE** to our comparison in Table 1. We give a more detailed analysis of the differences between their approach and ours in Appendix C.
> - Although wd1 achieves better results than d1, they rely on the same mean-field approximation and their main contribution is a slightly different objective that gets rid of the importance ratio. Thus we believe a comparison with this work is not necessarily as insightful as other comparisons.
> - DiffuCoder is pre-trained and post-trained for a very different set of tasks, i.e. coding instead of mathematical reasoning, which makes comparison with LLaDA + AGRPO difficult.
> - We also add LLaDA-1.5, which is trained using Variance-Reduced Policy Optimization (VRPO), and Diffusion Chain of Lateral Thought (DCoLT), which we believe are recent, more illuminating comparisons.

---

> > ### Author Response · Authors · 2025-11-28
> > **Response to Reviewer 3eva (cont.)**
> >
> > To address some of the reviewer's questions:
> > 1. **How do you generate intermediate masked states at time t?**
> >
> >     As described in the practical considerations section (which has been moved to Appendix E), we can efficienctly **cache** all intermediate states as the response is generated by marking each token with its unmasking step. Given the fully generated output, to sample the exact state at $t$, we can simply mask out all tokens that are marked with timestep $\\ge t$.
> >
> > 2. **What likelihood is used for the KL estimate?**
> >
> >     As described in previous responses, the KL is computed over action distributions where an action is a single denoising step, which means a) that it can be efficiently sampled via the same MC approach, and b) the likelihood used is the likelihood of the specific tokens that were unmasked at time t. In other words, we feed the intermediate state at time t ($o_t$) to both the on-policy and reference models and compare the probabilities of the tokens unmasked on that step ($o_{t+1}\mid o_t$).

---

### Author Response · Authors · 2025-11-28
**Summary of manuscript changes**

Thank you (and happy Thanksgiving) to all the reviewers for the thoughtful feedback and suggestions. We have made several major edits to the manuscript which should address many of reviewers' concerns and clear up any possible misconceptions.

### 1. Clearer framing of RL setup and contributions
- We believe there might have been some misunderstandings about the RL setup. These are reasonable given that readers might only be familiar with the traditional autoregressive setup in LLM RL papers, e.g. the GRPO paper. We want to emphasize that although we adopt the same objective form as GRPO, the inner sum is considered to be a sum over *timesteps* rather than a left-to-right sum over *token positions*, consistent with original policy gradient methods like REINFORCE/PPO.
- A core assumption we make is that post-training diffusion models should be treated as a *multi-step* MDP, where an action is a single unmasking step, rather than a *one-step* MDP where the model sees the prompt and generates a sequence in one action (which requires sequence likelihoods). For AR models, these are effectively the same due to left-to-right sequence factorization, so the one-step view makes sense. But for diffusion models, which lack a tractable sequence likelihood, the multi-step view is more natural [1] [2] and forms the basis of AGRPO.
    - Such a multi-step view means that in order to collect all m action likelihoods, one would naively need m model calls. This is where MC sampling comes in, a main contribution of AGRPO.
- We have **largely rewritten Section 3.1**, which talks about the MDP view, as well as parts of the abstract and introduction to better emphasize this distinction between AGRPO and prior dLLM works that try to fit the one-step view with heuristic approximations.

### 2. Stronger results/ablations
- Table 1 has been updated with more recent experimental results, including **multiple generation lengths**, as per 3eva's suggestion. We now include more suggested baselines as well (some of them only report results for single generation lengths).
    - Markers have been added to the table, e.g. for LoRA vs full fine-tuning, to help guide comparisons.
- Figure 3 has been updated with multiple lengths, and the surrounding phrasing has been changed to emphasize the novelty of these results.
- Reviewers noted that some claims made about compute/performance tradeoffs arising from the number of MC samples k were unsubstantiated. We added a new section 5.3 where we **ablate k**, studying the effect on training dynamics and wall time for GSM8K. These ablations confirm our hypothesis that scaling k does indeed improve fidelity, albeit at the cost of increased wall time (we note the exact relationship between compute/fidelity is complex and hard to quantify).

### 3. Clearer notation
- As suggested by 2DXg, the notation $o_{<t}$ has been dropped to avoid confusion with AR conventions. We now use $o_t$ to denote the intermediate masked state at time $t$. Explanations in Sections 3.3 and 4.1 have also been improved to ease readers into the diffusion-specific notation/objective. Specifically, as noted above, $t$ refers to a timestep index rather than a token position index.
- The KL estimator (Schulman, 2020) is now explicitly shown in Section 4.1.

### 4. Comparison to related methods
- Several suggested recent works on post-training dLLMs, i.e. PADRE, VRPO, and DCoLT [2], have been incorporated into the paper. We compare AGRPO to these methods both empirically, in the Results section, as well as more theoretically in the new Appendix C, which we hope will help reviewers better situate our paper's contributions.
- Section 3.2 has been updated to reflect earlier papers on RLHF with LLMs, which then evolved into RL/policy gradient methods for reasoning, as suggested by 9uBq.
- Although there is no official "Related works" section, we point out that Sections 2 and 3 are devoted to understanding the the current state of diffusion language models and RL post-training. Together with the comparisons for dLLM-specific methods in Appendix C, we believe these sections cover the role that a traditional "Related works" section would play.

[1] Black et al., 2024. Training diffusion models with reinforcement learning.

[2] Huang et al., 2025. Reinforcing the Diffusion Chain of Lateral Thought with Diffusion Language Models

---

### Meta-Review · Area_Chair_qBfy · 2025-12-22

**Summary:**

The current manuscript aims to tackle the problems in RL for diffusion LLMs. In particular, the authors argue that previous methods were based on unprincipled heuristicsm, and they then propose a new algorithm, Amortized Group Relative Policy Optimization (AGRPO), designed specifically for diffusion LLMs. The paper demonstrates the performance of AGRPO on three math/reasoning tasks, which shows the effectiveness of their method compared to previous methods such as diffu-GRPO.

However, while the current paper is considered an interesting exploration, concerns about the overall quality of the research work and paper writing were brought by the reviewers, including theoretical soundness and framing, experimental rigor and baselines, lack of ablation studies as well as presentation clarity. The authors revised the paper and provided the corresponding rebuttal comments that partially addressed the concerns from the reviewers. Although the current paper has been greatly improved in the rebuttal period, I would encourage a more significant major revision to make the current research to meet the high standard of ICLR.

**Reviewer Concerns:**

- Soundness of theoretical justification (Reviewer 3eva, 2DXg): Questions about lack of tractable sequence likelihood remains not fully addressed, although the authors argued this is not a critical flaw.
- Generalization beyond math tasks (Reviewer 9uBq): While the authors added more baselines, they did not expand the task variety (e.g., to code or general instruction following).
- Hardware efficiency comparisons (Reviewer 3eva): The authors provided $k$-sample ablations but didn't provide a granular FLOP-to-FLOP comparison against every baseline.

**Reviewer Scores:**

The authors' rebuttal should partially address each reviewer's concerns. Some up-rating should be expected, while the current paper is yet to be futher improved to get a consensus of accept.

---

### Decision · Program_Chairs · 2026-01-26

Reject